# Cross Dataset Analysis of Domain Shift in CXR Lung Region Detection

**DOI:** 10.3390/diagnostics13061068

**Published:** 2023-03-11

**Authors:** Zhiyun Xue, Feng Yang, Sivaramakrishnan Rajaraman, Ghada Zamzmi, Sameer Antani

**Affiliations:** Computational Health Research Branch, National Library of Medicine, National Institutes of Health, Bethesda, MD 20894, USA

**Keywords:** domain shift, lung region detection, chest X-ray datasets, catastrophic forgetting, modality-specific initialization

## Abstract

Domain shift is one of the key challenges affecting reliability in medical imaging-based machine learning predictions. It is of significant importance to investigate this issue to gain insights into its characteristics toward determining controllable parameters to minimize its impact. In this paper, we report our efforts on studying and analyzing domain shift in lung region detection in chest radiographs. We used five chest X-ray datasets, collected from different sources, which have manual markings of lung boundaries in order to conduct extensive experiments toward this goal. We compared the characteristics of these datasets from three aspects: information obtained from metadata or an image header, image appearance, and features extracted from a pretrained model. We carried out experiments to evaluate and compare model performances within each dataset and across datasets in four scenarios using different combinations of datasets. We proposed a new feature visualization method to provide explanations for the applied object detection network on the obtained quantitative results. We also examined chest X-ray modality-specific initialization, catastrophic forgetting, and model repeatability. We believe the observations and discussions presented in this work could help to shed some light on the importance of the analysis of training data for medical imaging machine learning research, and could provide valuable guidance for domain shift analysis.

## 1. Introduction

Chest radiography is an important imaging tool for the examination, identification, and diagnosis of cardiothoracic and pulmonary abnormalities. Radiological findings are frequently used for triage, screening, and diagnosis. The computer-aided diagnosis (CAD) of chest X-rays using deep learning (DL) and image processing techniques has been actively studied in the literature. A very recent comprehensive survey on publications using DL on chest radiographs can be found in [1]. However, despite this extensive research, very few methods have been translated into real-world clinical use.

Domain shift is a significant challenge that machine learning (ML) algorithms often face when models are deployed for real-world use. It refers to the phenomenon of unreliable prediction performance when the distribution of the data used to train and evaluate ML models in the development stage is different from that of the data seen by the deployed models. Because of the existence of domain shift, the performance of models during deployment may be significantly worse than what was observed during developmental experiments. This issue can be more substantial for medical imaging applications due to several factors: (i) training data size-medical images are often available either in small quantities, especially for abnormal cases; (ii) limited number of annotations due to the shortage of medical experts as well as the required intensity of labor efforts; (iii) lack of diversity in the distribution of patient population as data may be sourced from a single site; (iv) lack of variety in severity and type of disease manifestations; and (v) lack of multiple imaging modalities. Furthermore, the images obtained from different clinical providers are often taken by different imaging devices with varying manufacturer sensor designs and on-device post-processing, image acquisition parameters/protocols, and illumination conditions for optical imagery. As a result, the characteristics of images from different sources can be considerably different, which may create domain shift issues and low generalization performance of models on target. Therefore, it is of high value to study the problem of domain shift in medical applications and develop methods to provide necessary controls or, ideally, remedy it.

As summarized by [1], there have been limited works on domain adaptation for automated chest X-ray analysis [2,3,4]. In this work, we focus on an important pre-processing step in chest X-ray analysis—lung region detection to analyze domain shift problems in localizing lung region-of-interest (ROI). The problem of extracting the bounding box that encloses two lungs, as shown in Figure 1, can reduce the interference of irrelevant areas in the image for cardiopulmonary diseases and lessen the challenge of learning data-driven DL models in the succeeding steps, especially when the data are limited. 

To investigate the domain shift for lung region localization, we used five chest X-ray datasets. Each dataset contains images in a range of a few hundred or less. These datasets were collected from different sources, and they vary from each other in multiple aspects, including patient population, disease manifestation, imaging devices, clinical providers, and the number of images. DLs are data-driven. Therefore, the characteristics of data have a significant impact on the performance of DL models and play a key role in explaining and understanding the model behavior. Domain shift, also called distributional shift, in essence, is due to the changes in data characteristics. Hence, to obtain some insights into model explanation and analysis, we need to analyze and compare the data characteristics among these datasets first. We conducted the comparison and analysis from three aspects: information obtained from metadata or image header, image appearance, and feature extracted from a pretrained model. Through these three complementary approaches, we evaluated the datasets for homogeneity, diversity, and variability.

To examine the effect of domain shift on DL models, we carried out extensive experiments in four scenarios. We trained lung region detection models using individual datasets as well as combinations of datasets. We evaluated and compared the intra/inter-dataset performances among all models. We observed and discussed interesting results. We also experimented with modality-specific initialization, i.e., the model to be trained on one CXR dataset is initialized with the weights from the model that has been trained on another CXR dataset. We evaluated the additional effects that are often encountered in medical AI applications, viz., catastrophic forgetting and model repeatability.

It is very important to understand and explain the reasons behind such observations of performance variations, that is, why does a certain model work better than another model on a certain dataset? To this end, we proposed a new and simple approach that converts the multi-scale feature maps extracted from several stages in the applied object detection network into a feature vector, and generated feature embeddings in a 2D plot to show the feature representation characteristics of the network for images from different datasets.

To summarize, our main contributions include the following:We designed and carried out extensive experiments using five small chest X-ray datasets to study the cross-dataset performance and domain shift issue on lung detection in CXRs.We proposed to use three complementary approaches (at text, image, and feature level, respectively) for data analysis and understanding, a key prerequisite step that needs to be carried out before DL design and implementation but is often paid insufficient attention to in the literature.We considered and compared four scenarios in the experiments, using or not using the dataset combination, as well as different model initializations.We proposed a new method to extract and visualize the features from the object detection models which were shown to be helpful for providing insights into explaining the obtained detection performances on datasets.

Although our methods were developed and evaluated for lung region detection, a vital step in a CAD system for CXR analysis, they can be applied and adapted to other medical image analysis applications. We hope the observations and discussion of the experimental results presented in this work help shed some light on the importance of data analysis for medical imaging machine learning research, especially when the dataset at hand is small, and provide valuable input for domain shift analysis. In the following Section 2 and Section 3, we present detailed descriptions of the analysis and comparison of dataset characteristics, the methods for detecting lung ROI, the approach for analyzing domain shift across datasets, the design of the experimental tests, and the discussion of the results. We conclude the paper and provide suggestions for future work in Section 4.

## 2. Methods

### 2.1. Datasets

We used five deidentified chest X-ray datasets in this work that were collected from different sources and have manual lung masks: (1) Montgomery; (2) Shenzhen; (3) JSRT; (4) Pediatric; and (5) Indiana. The bounding boxes of manual lung masks were used as ground truth for this detection work.

Both Montgomery and Shenzhen datasets are made publicly available by the U.S. National Library of Medicine (NLM) [5]. The Montgomery set was sampled from images acquired by the Department of Health and Human Services, Montgomery County, Maryland, under its Tuberculosis (TB) Control program over many years. It consists of 138 posterior-anterior (PA) X-rays (80 controls and 58 TB cases with manifestations of tuberculosis), left and right lung lobe binary masks for each image, as well as patient age and gender information. The consensus annotations of regions of manifestations from two radiologists and their radiology readings were also added to the dataset later [6]. The Shenzhen set was collected and provided by Shenzhen No.3 Hospital in Shenzhen, Guangdong providence, China. It contains 326 normal chest X-rays and 336 abnormal chest X-rays showing various TB-consistent manifestations. The dataset also includes consensus annotations of regions of manifestations from two radiologists. The use and sharing of both the Montgomery and Shenzhen sets were reviewed and exempted from IRB review by the NIH Office of Human Research Protections Programs. The manual binary lung masks of a subset (566 images) of the Shenzhen dataset were provided through Kaggle by another research group [7]. JSRT [8] is a public chest radiograph dataset released by the Japanese Society of Radiological Technology (JSRT) two decades ago. There are 247 scanned chest radiographs in the dataset, 154 of which have malignant or benign nodules and 93 have normal lungs. The manual binary masks of the lungs for each chest X-ray are also available [9]. Associated textual information includes patient age, gender, nodule diagnosis, and coordinates of nodule location. The pediatric dataset was acquired from a private clinic in India. It contains 161 pediatric chest radiographs. Each image has a corresponding manual lung segmentation mask. We also used a very small subset (55 frontal images) of the Indiana University hospital network image collection, made available through Open-i [10], that have manual lung masks. Each image in this subset has a clinical report that included information on findings and impressions but no patient demographic information such as age and gender.

#### Pre-Processing

The image formats in the five datasets may be different from set to set, although PNG and DICOM are two of the main formats. For example, the images in the Pediatric dataset are in both formats, where the PNG images have 12-bit gray-scale color depth, while the JSRT has PNG images of 12-bit gray-scale (also converted to TIF images of 8-bit gray-scale [11]) and the PNG images in Indiana set are with 8-bit gray scale (were contrast-enhanced for the convenience of lung mask lineation). The ground truth lung segmentation mask images may be in TIF, GIF, and PNG formats, respectively. We converted the images of all the datasets to the JPG format of 8-bit gray scale. It should be noted that special attention needs to be given when converting images [11]. We generated the ground truth lung region bounding boxes from the manual lung segmentation masks provided in each dataset. For the Shenzhen dataset, since only 566 images have lung segmentation masks available, we manually drew the bounding boxes of lungs for all the remaining 96 images (using the Matlab ImageLaber tool). The dimensions of images vary across datasets and within some individual datasets. For example, the Montgomery dataset has two distinctive sizes (4020 × 4892 or 4892 × 4020 pixels), and the Pediatric dataset images are in varied resolutions (2446 × 2010, 1772 × 1430, and 2010 × 1572 pixels). The lung masks may be of different sizes to the corresponding images. We resized all the images (and corresponding masks) to be on the same scale. In addition to the whole images, we also generated the so-called cropped lung images where the images were cropped to the lung region box.

### 2.2. Data Analysis and Comparison

The DL models are data-driven such that their performance can be significantly influenced by their data characteristics. However, the robustness, reliability, and accuracy of models can be improved through better DL architecture design, hyperparameter optimization, and training strategy. Therefore, the step of analyzing the training data itself is very important and can provide valuable information and insights toward robust and effective DL algorithm design, implementation, and evaluation. As a result, we first examined and compared the data characteristics among the datasets at three levels: text, image, and feature.

#### 2.2.1. Analysis of Textual Information Obtained from Metadata or Image Header

These datasets vary from each other with respect to geographical regions, populations, diseases, imaging devices, providers, views, dataset size, image formats, image size, and gray scale depth. A summary of the information on these aspects of each dataset is provided in Table 1. We extracted some text information from the DICOM header if the dataset did not directly provide related information in their description or through the papers. We put “N/A” in the table if we did not find relevant information. Since all the images input to the DL network were resized to have the same scale in dimension and their intensity pixel values were converted to have the same depth (8-bit) in a JPG image format, these three attributes of intensity depth, image dimension, and image format were not included in the table. For easy visual comparison, we generated the pie charts with respect to the ratio of disease, gender, and age category in the datasets in which such information is available. They are displayed in Figure 2, respectively.

#### 2.2.2. Analysis of Image Appearance

Besides comparing the datasets using textual information, comparing images themselves in different datasets is also highly desirable as they are the data directly input to and used by the DL networks. Although manually browsing the images in each dataset can help to provide some extent of understanding and perception of what the images in each dataset look like, it is appealing and vital to have a general representative picture that can show the characteristics of the images in each set (at least to some degree) as it can be perceived promptly. To this end, we used a simple approach which was to create the average image of the whole images [12], as well as the cropped lung images of each set. This approach was carried out by finding the mean width and the mean height of all images first, then resizing all images to have the width and the height equal to the calculated mean width and mean height, respectively, and then adding the resized images all together and taking the average value at each pixel. Figure 3 shows the average whole image and the average cropped lung image calculated from each dataset, respectively. As demonstrated by Figure 3, the shape, intensity, and size of the lung areas as well as the whole upper body are different from one average image to another; although, there are similarities due to the intrinsic anatomical structure of body and organs. Among the five datasets, the average image of the Pediatric dataset is the most distinguished from that of other datasets regarding body and lung shapes, which is consistent with clinical observations [13].

#### 2.2.3. Analysis of Features Extracted from the Pretrained Model

To obtain more insight into the lung region differences across datasets, we also extracted the feature vectors from the whole images as well as the cropped lung region images in each dataset using a DL classification network and visualized those features in a 2D space. For the DL classification model, we used an ImageNet trained Swin Transformer. Swin Transformer [14] was developed by aiming to make the transformer architecture designed originally for Natural Language Processing (NLP) more suitable for vision applications. It constructs hierarchical feature maps based on the key idea of utilizing shifted window partitioning for calculating self-attention locally, and achieves linear computational complexity w. r. t. image size. In our work, we used the Swin-B model which uses 384 × 384 pixels as input image size, 4 × 4 pixels as patch size, and 12 × 12 pixels as window size. The feature vector at the average pooling layer before the classification head layer was extracted. The feature has a length of 1024. For dimension reduction and feature visualization, we used UMAP (Uniform Manifold Approximation and Projection) [15]. UMAP, like tSNE, generates a low-dimensional graph which is optimized to be as structurally similar as possible to the high-dimensional graph representation of the data it has constructed, but may be faster and may preserve global structure better [15]. The UMAP plots of the ImageNet Swin-B model features of the five datasets obtained from using the whole image as the model input are displayed in Figure 4a. The whole image features of the five datasets are separated very well from each other (with the exception of only a few images from one dataset falling in the cluster of another dataset). We also extracted the same types of features for cropped images. As seen in its UMAP plot, shown in Figure 4b, the cropped image features among Montgomery, JSRT, Pediatric, and Shenzhen sets are well separated from each other, but the Indiana cluster blends with the Shenzhen cluster, indicating these two datasets have high similarity w. r. t. this specific type of features. Another observation is that there is a small number of Shenzhen images that are closer to the Pediatric cluster. We checked these Shenzhen images and found that they are pediatric images contained in the Shenzhen set. Although it should be noted that observations are dependent on what specific kind of features are used for analysis, they demonstrate that there are differences existing between the images in these datasets to a degree.

### 2.3. Lung ROI Detection Network

Object detection networks can be generally categorized into two types: one-stage detectors and two-stage detectors. One-stage detectors omit the step of region candidate proposal, a key component in two-stage detectors, and have object classification and bounding box regression performed directly using anchors extracted from the feature maps obtained from the entire image. Representative detection networks include Faster RCNN [16], YOLO [17], RetinaNet [18], SSD [19], DETR [20], etc. For a comprehensive literature review of object detection networks, please refer to a very recent survey paper in [21]. To localize the lung ROI, we applied a recent variant in the one-stage detector family of YOLO algorithms, that is, YOLOv5 [22]. Since the proposal of the original network version in 2016, YOLO has gone through multiple versions with various changes and improvements regarding backbone network, loss function, feature aggregation, data augmentation, activation function, normalization methods, regularization methods, optimization methods, among others [23]. On the shoulders of previous versions of YOLO (v1–v4), YOLOv5 was developed in Pytorch and is available in GitHub as an open-source package [22]. It is actively maintained and constantly improved by Ultralytics. Innovative and practical engineering maneuvers as well as algorithm bells and whistles have been applied, implemented, added, and adapted regularly. YOLOv5 itself has four variants of model structures that have different memory storage sizes. The general architecture of YOLOv5 models consists of three modules: (1) backbone—for extracting features of various sizes from the input image; (2) neck—for generating feature pyramids and performing feature fusion; and (3) head—for performing the final detection which consists of both bounding box regression and class prediction. The specific model structures, training strategies, loss functions, augmentation methods, as well as other up-to-date implementation and algorithm details of YOLOv5 can be found in its repository [22].

### 2.4. Feature Visualization of Lung ROI Detection Network

Besides evaluating the detection performance within a single dataset or across different datasets, we were also interested in understanding why a certain detection model works well/better on a certain dataset but not on another dataset, and explaining the generalization discrepancy across datasets. To this end, we proposed a new method for analyzing the features extracted from the YOLOv5 network. Different from other detection networks that contain fully connected layers, such as Faster RCNN, the YOLOv5 network consists of convolutional layers whose outputs are three-dimensional feature maps before the head module. To generate UMAP plots, as shown in Figure 4, which require the use of feature vectors, we first selected the three groups of feature maps (having different scales) that are the inputs to the head module in the network. Then, we applied the global average pooling to the feature maps in each group by which those feature maps in each group were converted into one feature vector. Next, we concatenated the feature vectors of all three groups to generate the final feature vector for each image. Last, we used the feature vectors extracted from all the images of interest to create a corresponding UMAP plot. Based on our best knowledge, there is no such work reported in the literature on generating feature visualization 2D plots for YOLOv5 models.

## 3. Experimental Results and Discussion

### 3.1. Experiment Settings

We split each dataset randomly (at the patient level) into training, validation, and test sets using a ratio of 70/10/20. The specific number of images in each set of each dataset is listed in Table 2, respectively. As shown in Table 2, except for the Shenzhen dataset which has the largest number of images with 463 in the training set, the size of the training set is quite small for all the other datasets, especially the Indiana dataset.

To alleviate over-fitting, we used the YOLOv5s model structure (which has the smallest storage size among the four YOLOv5 structures) and initialized the weights using a COCO pretrained model. YOLOv5 also utilizes several types of image augmentation such as color modification, scaling, translating, flipping, and mosaic augmentation. Mosaic augmentation, a novel augmentation method proposed by YOLOv5, generates a new training image that consists of four tiles with a random ratio obtained by combining one original image and three other randomly selected images. The specific software version setting we used was YOLOv5s 6.0. The backbone, neck, and head parts of its model structure are based on CSP-Darknet53 [24], SPPF [22] and PAnet [25], and YOLOv3 head layers, respectively. It uses binary cross entropy loss for calculating both classification loss and objectiveness loss, and CIoU [26] loss for computing bounding box regression loss. A summary of information on this specific version including employed training strategies can be found in [27]. For training, the batch size was 16, the number of epochs was 100, and the input image size was 640 × 640 pixels. For other hyperparameters and arguments (such as optimizer, initial learning rate, momentum, weight decay, warmup epochs, augmentation methods, etc.), the default values were used. For testing and evaluating, we set the image size as the same as that in training, the confidence threshold to be 0.25, the IoU threshold to be 0.45, the maximum number of output detections to be 1, and kept the other parameters to be the same as default values. The models were trained on a Lambda server with 8 GeForce RTX 2080 Ti GPUs. Unless specifically pointed out, the parameters, software (dependency library versions) and hardware settings remained the same for all the experiments presented and discussed in this paper.

For feature extraction, we converted the multi-scale feature maps at the stage 23, 20, and 17 of the YOLOv5 model, respectively, into a feature vector using global average pooling, and then concatenated the feature vectors obtained from the three stages (with the order of stage 23, 20, and 17). For example, at the stage 23, the global average pooling takes the average of the 17 × 20 feature map at each of the 512 channels and outputs a feature vector with a length of 512. The final feature vector obtained by concatenating feature vectors of all three stages has a length of 896 (=512 + 256 + 128). The feature vectors of all the images of interest were extracted from a model and then used to generate a UMAP plot.

### 3.2. Experiments in Four Scenarios

To investigate and analyze domain shift across datasets, we considered and carried out experiments in the following scenarios: (1) models trained using each individual dataset; (2) models trained using a combination of all the datasets except one; and (3) models trained using a combination of all the datasets. For all the above three scenarios, the models were initialized using the weights of the model pretrained with the COCO dataset. To check and verify the existence and extent of catastrophic forgetting and the effectiveness of modality-specific initialization, we also examined another scenario: (4) models trained using each individual dataset but initialized using weights from another model that was trained with a different dataset. Figure 5 shows the example workflow diagrams in the above four scenarios, respectively. All the models were evaluated and compared using the test set of each individual dataset. We used mAP@0.5:0.95 as the evaluation metric.

To check the repeatability of the model performance (i.e., to see if the model produces the same result for the same experiment and the same setup), we re-trained some models several times on the same GPU server using the same training/validation set, while keeping the settings of software environment, network hyperparameters, code version, and arguments the same, and examined testing performance on the same test set. The performances of all runs in each repeatability experiment were observed to remain the same. Table 3 lists the test performance of models trained in the first three scenarios and Table 4 lists the test performance of models trained in the fourth scenario.

#### 3.2.1. Scenario 1: Models Trained Using Each Individual Dataset

The Scenario 1 section in Table 3 shows the results of testing the model trained with each individual dataset on the test set of each of the five datasets. The mAP@0.5:0.95 values in this section indicate that the within-dataset performance is higher than any of its cross-dataset performances for all the models except the Shenzhen model. For example, the JSRT model (the third row in the Scenario 1 section) achieves 0.964 on its own test set but 0.583, 0.723, 0.817, and 0.883 on the test sets of Indiana, Pediatric, Montgomery, and Shenzhen datasets, respectively. The cross-dataset performances of these individual dataset models vary considerably from model to model on the same test set. The Shenzhen model obtained the best cross-dataset performance among all models. It is the second-best performing model for the test set of Pediatric, JSRT, and Indiana datasets, and is even significantly better than the Montgomery model on the Montgomery test set. We hypothesize that one key factor contributing to this performance gain is that the size of Shenzhen dataset is significantly larger than that of the other four datasets. As a result of having a larger volume, its data diversity can also be increased, which boosts its chance to better represent the data of other datasets and reduce the extent of domain shift. This hypothesis seems to be backed up by UMAP plots in Appendix A Figure A1 (showing features of images in all test sets extracted from these individual dataset models). Except the UMAP plot for the Shenzhen model (Appendix A Figure A1b) where the feature cluster of the Shenzhen training set seems to be mixing well or close with that of test images in the other four datasets, the features of the training dataset in all the other four UMAPs (Appendix A Figure A1a,c–e) look generally well separated from those of test datasets, unless the test set is from the same dataset as the training set. The UMAP plots in Appendix A
Figure A1 can also shed some light on why a certain model performs significantly worse on a certain dataset. For example, for the JSRT model (Appendix A Figure A1c), the features of Pediatric and Indiana test images are far from those of the JSRT training/test images. This observation aligns with the detection performance comparison between different test datasets for this model, which is indicated by the mAP@0.5:0.95 values in Table 3. The agreement between the observations from these UMAPs and the quantitative evaluation results demonstrates the usefulness of the proposed YOLOv5 feature analysis method.

#### 3.2.2. Scenario 2: Models Trained Using All Datasets but One

One approach that can reduce domain shift issues across datasets is to combine the labeled training images from all available sources. It is based on the expectation that the data from different sources may be complementary to each other and by combining them, the diversity of source data can be increased, which, in turn, could lead to the improvement in the feature representation capability and the network generalization ability with respect to the data distribution in the target domain. In Scenario 2, we wanted to examine the performance of models trained using all datasets except one, especially on the dataset that was excluded from the training process. As shown in the Scenario 2 section of Table 3, the cross-domain performance was indeed substantially improved for all the models with this simple approach of combining datasets. For example, the “All-Pediatric” model (trained with the combination of all datasets but the Pediatric dataset) achieved 0.903 on the Pediatric test set, while all the other non-Pediatric individual models (shown in the “Pediatric” column of the Scenario 1 section in the Table 3) obtained 0.681, 0.889, 0.723, and 0.777, respectively, on the same test set. Appendix A Figure A2 displays the UMAP plots for each model in Scenario 2, where the features of images from the training datasets and the target test dataset are visualized. For example, regarding the All-Montgomery model, the embeddings of features from the Shenzhen, JSRT, Pediatric, and Indiana training sets and the features from the Montgomery test set are shown in Appendix A Figure A2a with different colors, respectively. It can be observed that for the same test set, the feature space of training images covers that of test images much better than that of the individual training set. This demonstrates the effectiveness of combining training datasets that are obtained from different sources for our specific datasets and task, even though the combined training dataset does not contain any images from the target source. As shown by comparison to Appendix A Figure A1, in general, the feature space of training images in Appendix A Figure A2 becomes more spread, has a larger overlapping area with, or is closer to that of, test images due to the increase in data volume and diversity. Besides the significant improvement on cross-domain prediction performance, the models remain working well on the test images that are from the same source as the training images. For example, the mAP0.5:0.95 values of the All-Indiana model are 0.987, 0.954, 0.986, 0.953 on Montgomery, Shenzhen, JSRT, and Pediatric test sets, respectively. Similarly, by comparing the values in each column in the Scenario 2 section in Table 3, we can observe that among all five All-1 models, the models trained with images including those from the target dataset have either better or comparable performance than the model trained without such images.

#### 3.2.3. Scenario 3: Model Trained Using All Datasets

In this scenario, we trained a model using the combination of training images from all five datasets. We then checked and evaluated its performance on each dataset’s test set. The results are given in the last row of Table 3. As expected, including the training images from the target dataset increases model performance on the target dataset. That is, the mAP0.5:0.95 value in each column of the last row (for the All model) is significantly larger than that in the diagonal line of the Scenario 2 section in Table 3. For example, the performance on the JSRT test set is improved from 0.954 (All-JSRT model) to 0.978 (All model) by adding JSRT training images to the training set. One interesting observation exhibited by comparing the results of Scenario 3 and Scenario 2 is that although increasing data volume and diversity increases the chance of making the source domain data represent the target domain data better, it may not always be the case. For example, for the Montgomery test set, the All model is markedly outperformed by the All-Pediatric model and the All-Indiana model (0.976 vs. 0.987 and 0.987). It indicates, for this case, that using training images from four of the five datasets can produce better performance than using images from all five datasets. Therefore, adding more data may not necessarily produce better results and alleviate domain shift issues, even if the quality of the added data is good. The characteristics of the data to be added and how similar it is to that of the target domain play an important role as well. We tried to see if we could obtain some explanations and insights for this phenomenon by comparing the UMAPs of these models (Appendix A Figure A3b–d), but it seems that we are unable to draw a conclusive decision from those UMAPs regarding it. This demonstrates the challenges of explanation and analysis for network prediction, as well as the complicated factors contributing to network generalization capability, signifying the need to make more efforts on such kinds of research and experimental evaluations.

#### 3.2.4. Scenario 4: Models with Modality-Specific Initialization

To train a deep network with a small medical dataset, one commonly used technique is applying transfer learning, that is, initializing the model architecture with weights from a model pretrained with a huge dataset, such as ImageNet. Recently, there have been studies showing that using modality-specific initialization, that is, a model pretrained with the same modality of medical images (with annotations from a task different from the one at hand), can produce better performance for the medical imaging applications than the one pretrained with the frequently used general-domain image dataset (ImageNet) [28]. In this experiment scenario, we were interested in checking if this observation holds when the model is initialized, using the weights of models trained with a different dataset in our five datasets. Such experiments also allow us to examine another issue caused by the existence of domain shift-catastrophic forgetting, i.e., the model forgets what it has learned from the previous dataset after fine-tuning on the new dataset. Table 4 lists the performance comparison of each individual model that was trained using different initialization weights. For example, in the sub-section of the Shenzhen model in Table 4, the first column lists the name of the pretrained model, the second column shows the performance on the Shenzhen test set of the model fine-tuned with Shenzhen training set, and the third column shows the performance of the fine-tuned model on the test set of the dataset that was used in the pretraining. From Table 4, we observed that only for the Montgomery model, the modality-specific initialization with any of the five pretrained models (Shenzhen, JSRT, Pediatric, Indiana, and All-Montgomery) outperforms the general-image initialization (Yolo5s which was trained with COCO dataset) considerably. For the other four individual dataset models, using modality-specific pretrained models is not always beneficial. For example, for the Indiana model, using the model pretrained with the Pediatric dataset performed markedly worse (0.903) than that with Yolo5s (0.938), while using the Shenzhen pretrained model accomplished a significant gain in performance (0.964). We also noticed that using the All-1 pretrained model to initialize the model achieved better results than using the Yolo5s for all the individual dataset models, suggesting the modality-specific initialization can be of an advantage when using a larger dataset with more variety and diversity. By comparing the third column in each sub-section in Table 4 (performance of models on the old dataset after fine-tuning on a new dataset) and the diagonal value of the Scenario 1 section in Table 3 (original performance of the models on the same dataset before fine-tuning), we found that there was forgetting for all the individual dataset models except the Montgomery model fine-tuned with the Shenzhen dataset (the fine-tuned model obtained 0.956 on Montgomery test set, while the original model obtained 0.908).

### 3.3. Discussion

Data characteristics can have a great impact on the design and prediction performance of ML algorithms, especially for medical applications. The key characteristics for medical data include *Volume*, *Veracity*, *Validity*, *Variety*, and *Velocity* [29] which refer to the amount of data, the truthfulness of data, the quality and consistency of data, the diversity of data, and the generation duration of data, respectively. Analyzing these data characteristics can not only facilitate ML researchers to develop better and more suitable architectures/methods for the goal of increasing model reliability and robustness, but also help to obtain more information from data which is also of value to clinicians. Given that the volume of medical image data (especially labeled data) is generally small and image data from different clinical centers are usually different, it is important to investigate and examine the domain shift issue across small datasets from different sources. The presented work is mainly related to the study of data volume and variety, and their impact on domain shift for the specific task of lung region detection in X-ray images. Our analysis of the data in the five datasets indicated the existence of cross-dataset differences exhibited in image appearance due to multiple contributing factors, such as variabilities in sensors, populations, disease manifestations, on-device processing, and imaging conditions. It is desirable that ML models can tolerate the data variability across different clinical centers well and be reliable when deployed in a new center, even though no data from the new center were available in the training stage of the models. Generally speaking, it is expected that increasing the volume and variety of the training data will reduce domain shift and increase the reliability of ML generalization in an unseen environment. However, our experimental results revealed that it may not always be the case when having notably limited data. That is, adding more data may not necessarily produce better results even if the data have good quality. It also depends on the characteristics of the added data and their similarity to those of the target domain. Similarly, the benefit of using modality-specific pretrained models over the ones pretrained with the frequently used general-domain image dataset is not observed for some models, although the modality-specific initialization can be of an advantage when using a larger dataset with more variety and diversity. Therefore, special attention and caution need to be paid when utilizing these techniques to mitigate domain shift issues among limited data. It is of great help, especially for high-risk situations such as clinical applications, to have effective tools that can predict and analyze the likely behaviors of models in the target domain. To this end, we developed a method to visualize the model features which can show the difference of data distributions. It can explain model behavior to a certain extent. However, it has limitations, as it cannot produce conclusive analysis results for some model predictions. In this work, we focused on studying the domain shift issue for lung region detection without specifically considering the normality or abnormality of lungs, an initial effort toward building a reliable and robust CXR AI system. In the future, we will expand the work to evaluate disease detection which would have more clinical impact and attract more interest. Nonetheless, our experiments demonstrate that modal behaviors and performance can be different from common expectations when datasets are small, and our analysis methods can be applied to other medical imaging applications.

## 4. Conclusions

One of the ML challenges in medical image analysis is domain shift. That is, the data distribution of the training dataset is different from that of the test dataset, which may lead to significant performance degradation of ML models in a real-world deployment. In this work, we aimed to study and analyze the domain shift issue across multiple datasets w. r. t. the task of detecting lung regions. Lung region detection in chest radiographs is an important early step in the ML pipeline for pulmonary disease screening and diagnosis. Like many other medical imaging applications, manual annotations of lung regions are limited. We had gathered five small such datasets that were collected from different sources. Using these datasets, we made efforts from several aspects in order to study domain shift issue. Specifically, we proposed to examine the characteristics of the datasets and their differences from three levels: text, image, and feature. We compared the information extracted from metadata, created an average image for each dataset, and checked features extracted using a pretrained CNN classifier. To evaluate and compare model performance under different situations, we designed four experimental scenarios including training with an individual dataset as well as a combination of multiple datasets. We also checked modality-specific initialization, catastrophic forgetting, and model repeatability. In addition, we developed a new visualization method for the applied detection network to obtain an explanation on the model performance variations. We found that there was generally a good alignment among feature distributions in the 2D plots and the obtained values of metrics for quantitatively evaluating the detection performance of different models. This demonstrates the usefulness of the proposed visualization method, although some observations cannot be explained by the feature visualization plots. We discussed the observations from the experimental results which demonstrate the complicated nature of both domain shift and the effects of data characteristics on model capacity for small datasets. From the experimental results, we noticed two key observations: (1) although increasing data volume and diversity increases the chance of making the source domain data representing the target domain data better, it may not always be the case; (2) using modality-specific pretrained models may not always be beneficial. The insights, analysis, and observations provided by our work can be valuable for the understanding and alleviation of domain shift in medical imaging applications in which a small amount of data are available from each of the different sources. In the future, we will explore techniques in semi-supervised learning and active learning to remedy the domain shift for lung region detection, and extend the current work and analysis for abnormality detection in the lungs.

## Figures and Tables

**Figure 1 diagnostics-13-01068-f001:**
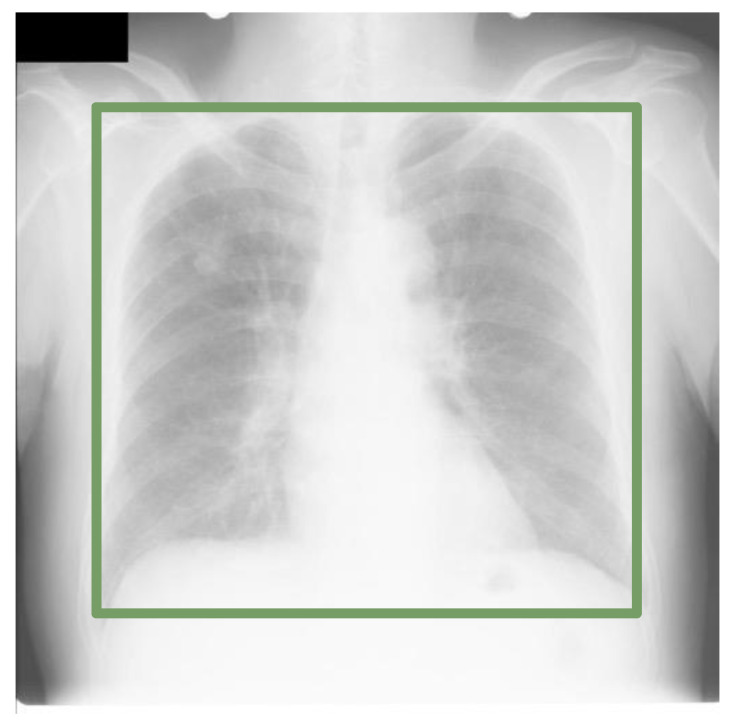
Lung ROI detection.

**Figure 2 diagnostics-13-01068-f002:**
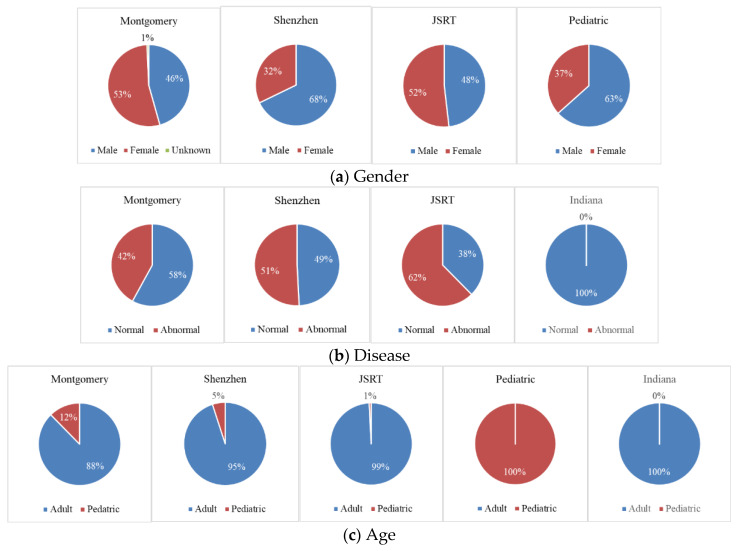
Comparison of datasets with respect to the percentage of gender, age, and disease categories.

**Figure 3 diagnostics-13-01068-f003:**
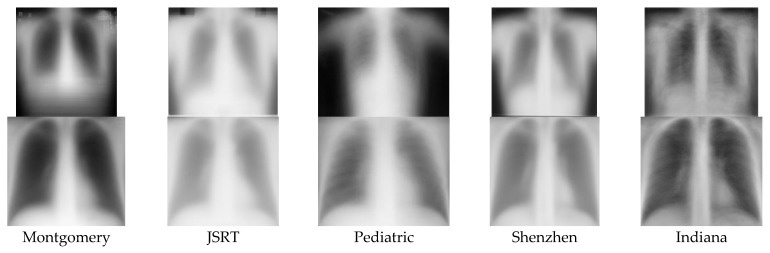
Average image of each dataset (1st row: whole image; 2nd row: cropped image).

**Figure 4 diagnostics-13-01068-f004:**
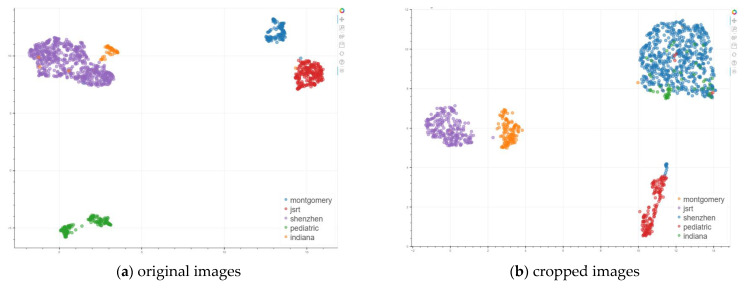
UMAPs of ImageNet Swin-B classification model features extracted from (**a**) original and (**b**) cropped images.

**Figure 5 diagnostics-13-01068-f005:**
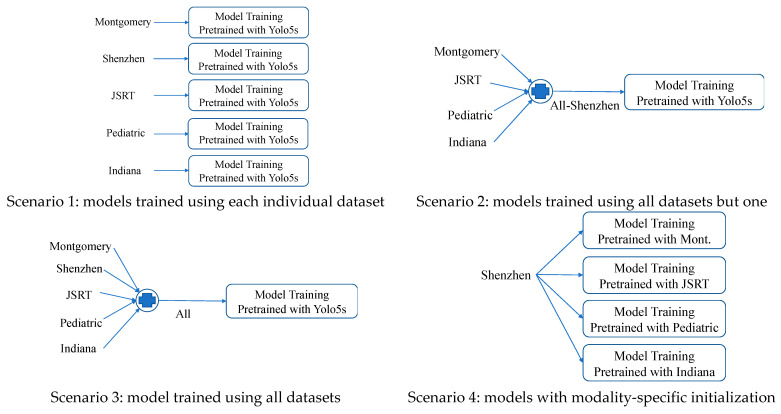
Diagrams of four experiments scenarios.

**Table 1 diagnostics-13-01068-t001:** Comparison of datasets.

Dataset	No. of Images	Disease	Country	Gender	Age	Device	View
Montgomery	138	Normal/TB	USA	Male, Female	Adult, Pediatric	Konica Minolta	PA
Shenzhen	662	Normal/TB	China	Male, Female	Adult, Pediatric	N/A	PA, AP
JSRT	247	With/without Nodule	Japan	Male, Female	Adult, Pediatric	Konica LD 4500 & 5500	N/A
Pediatric	161	N/A	India	Male, Female	Pediatric	Konica Minolta	PA, AP
Indiana	55	Normal	USA	N/A	Adult	N/A	PA, AP

**Table 2 diagnostics-13-01068-t002:** The number of images in the training/validation/test set in each dataset.

Datasets	Training	Validation	Test	Total
Montgomery	97	14	27	138
Shenzhen	463	67	132	662
JSRT	173	25	49	247
Pediatric	113	16	32	161
Indiana	39	6	10	55
Total	1263	885	128	250

**Table 3 diagnostics-13-01068-t003:** The performance (mAP0.5:0.95) of models on the test set of each individual dataset for scenarios 1–3.

Model	Test Set (mAP0.5:0.95)				
	Mont.	Shenzhen	JSRT	Pediatric	Indiana
Scenario 1: models trained using each individual dataset					
Mont.	0.908	0.875	0.500	0.681	0.813
Shenzhen	0.953	0.953	0.938	0.889	0.925
JSRT	0.817	0.883	0.964	0.723	0.583
Pediatric	0.829	0.872	0.451	0.904	0.831
Indiana	0.883	0.847	0.425	0.777	0.938
Scenario 2: models trained using all datasets but one					
All-Mont.	0.964	0.959	0.982	0.939	0.941
All-Shenzhen	0.963	0.946	0.977	0.943	0.953
All-JSRT	0.980	0.960	0.954	0.937	0.940
All-Ped.	0.987	0.959	0.990	0.903	0.979
All-Indiana	0.987	0.954	0.986	0.953	0.936
Scenario 3: model trained using all datasets					
All	0.976	0.958	0.978	0.932	0.955

**Table 4 diagnostics-13-01068-t004:** Trained with individual dataset but initialized with weights from models trained with another dataset.

Montgomery Model and Test Set			Shenzhen Model and Test Set		
Pretrained Model	Test on Montgomery	Test on the Dataset Used in the Pretrained Model	Pretrained Model	Test on Shenzhen	Test on the Dataset Used in the Pretrained Model
Yolo5s	0.908		Yolo5s	0.953	
Shenzhen	0.980	0.943	Montgomery	0.953	0.956
JSRT	0.943	0.938	JSRT	0.951	0.921
Pediatric	0.948	0.899	Pediatric	0.951	0.891
Indiana	0.965	0.893	Indiana	0.957	0.886
All-Montgomery	0.979	0.941	All-Shenzhen	0.958	0.939
**JSRT model and test set**			**Pediatric model and test set**		
Pretrained model	Test on JSRT	Test on the dataset used in the pretrained model	Pretrained model	Test on Pediatric	Test on the dataset used in the pretrained model
Yolo5s	0.964		Yolo5s	0.904	
Montgomery	0.945	0.742	Montgomery	0.896	0.852
Shenzhen	0.976	0.929	Shenzhen	0.926	0.931
Pediatric	0.946	0.819	JSRT	0.921	0.806
Indiana	0.961	0.586	Indiana	0.929	0.815
All-JSRT	0.987	0.934	All-Pediatric	0.930	0.939
**Indiana model and test set**					
Pretrained model	Test on Indiana	Test on the dataset used in the pretrained model			
Yolo5s	0.938				
Montgomery	0.931	0.897			
Shenzhen	0.964	0.926			
JSRT	0.908	0.709			
Pediatric	0.903	0.844			
All-Indiana	0.977	0.933			

## Data Availability

Montgomery dataset: https://data.lhncbc.nlm.nih.gov/public/Tuberculosis-Chest-X-ray-Datasets/Montgomery-County-CXR-Set/MontgomerySet/index.html (images and lung masks) (accessed on 6 March 2023)**.** Shenzhen dataset: https://data.lhncbc.nlm.nih.gov/public/Tuberculosis-Chest-X-ray-Datasets/Shenzhen-Hospital-CXR-Set/index.html (images) (accessed on 27 January 2023). https://www.kaggle.com/datasets/yoctoman/shcxr-lung-mask (lung masks) (accessed on 27 January 2023). JSRT dataset: http://db.jsrt.or.jp/eng.php (accessed on 27 January 2023)**.** Pediatric dataset: cannot be made public due to patient privacy constraints. Indiana lung mask subset: https://lhncbc.nlm.nih.gov/LHC-downloads/downloads.html (images and lung masks) (accessed on 6 March 2023).

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
