# Peer review of "Cross Dataset Analysis of Domain Shift in CXR Lung Region Detection"

_diagnostics, 2023, doi:10.3390/diagnostics13061068_

Round 1

Reviewer 1 Report

Thank you so much for giving me an opportunity to review this paper. It is a well-conducted and well-written manuscript. However, there are a few things that need to be addressed before considering it for publication.

1. Table 3: Train or training? Val, please provide the full form.

2.  Table 4: It isn't very clear. Please explain it properly.

3. There is no discussion for this article. Please provide it.

4. Please provide a clinical aspect of this study. 

Reviewer 2 Report

In this paper, the Authors aim to analyze domain shift in lung region detection in chest radiograph by using five chest X-ray datasets. They compared the characteristics of these datasets according to the information obtained from metadata, the image appearance and the features extracted from a pretrained neural network, and proposed a feature visualization method to provide explanations for the applied object detection model on the obtained results. Overall the paper is technically sound and the novelty stands clearly. I have only minor suggestions to point out, as follows:

1.     In the last part of the Introduction, sentences that are actually conclusions should be avoided and better stressed in the proper section.

2.     The unit of measurement for the resolution (i.e., pixels or mm) is missing in the entire text.

3.     I suggest adding an overall workflow (or one for each addressed scenario) to graphically display the pursued methodology.

4.     Conclusions should be strengthened.

Reviewer 3 Report

This is an interesting study dealing with a major issue in the field of computer aided deep learning tools in medical imaging (i.e., domain shift), which makes the translation of these tools in clinical routine difficult. 

The authors carried out an extensive experiment using different datasets of chest xRays to evaluate domain shift issue in the detection of ROI (lung detection).

They also proposed a fancy method to extract and visualise features used for lung detection for each DL model.

The findings of this study were expected since it's well known that the data volume and diversity are key elements for generalizable DL models. However, the authors showed that this is not always the case and there is still room for improvement using novel methods and developments. I think that this study is of great interest to understand issues that doctors and researchers might encounter in the generalization of DL models in clinical routine.

I do have two comments though:

- It could be interesting to evaluate disease detection too, rather than only normal structure detection (i.e., lungs)

- Is it possible to compare the trained models performances between normal chest xRays and pathological chest xRays ?

Minor comment: may be the manuscript should be shortened for better clearness
